# Length-of-stay and factors associated with early discharge after birth in health facilities in Guinea by mode of birth: Secondary analysis of Demographic and Health Survey 2018

Aline Semaan[1]*, Fassou Mathias Grovogui[2,3], Thérèse Delvaux[1], Natasha Housseine[4,5], Thomas van den Akker[6,7], Alexandre Delamou[2,3], Lenka Beňová[1]

1 Department of Public Health, Institute of Tropical Medicine, Antwerp, Belgium, 2 Centre National de Formation et de Recherche en Santé Rurale (CNFRSR) de Maférinyah, Forécariah, Guinea, 3 Africa Center of Excellence for the Prevention and Control of Communicable Diseases (CEA-PCMT), University Gamal Abdel Nasser, Conakry, Guinea, 4 Global Health Section, Department of Public Health, University of Copenhagen, Copenhagen, Denmark, 5 Medical College East Africa, Aga Khan University, Dar es Salaam, Tanzania, 6 Athena Institute, Faculty of Science, Vrije Universiteit Amsterdam, Amsterdam, The Netherlands, 7 Department of Obstetrics and Gynaecology, Leiden University Medical Centre, Leiden, The Netherlands

* asemaan@itg.be

**Data Availability Statement:** This paper used secondary data that are available on the DHS

## Abstract

The immediate postpartum period (first 24 hours after birth) represents a critical time for women and newborns. Postnatal length-of-stay varies globally; in Guinea, a 24-hour facility stay following childbirth is recommended, with an emphasis on providing frequent monitoring of mother and newborn for the first 6 hours. This study describes postpartum length-of-stay following facility-based births in Guinea, and investigates factors associated with early discharge. This cross-sectional study analysed secondary Demographic and Health Survey data covering the most recent livebirths during 2013–2018. We included 2,763 women who gave birth vaginally or by caesarean section in healthcare facilities. Early discharge following vaginal birth was defined according to two cut-offs (<24 hours and <6 hours); early discharge following caesarean section was defined as <72 hours. We assessed socio-demographic, obstetric and health-system factors associated with early discharge using binary and multi-variable logistic regression. Among women with a vaginal birth, 81.5% were discharged <6 hours, with a median length-of-stay of 3 hours. 28% of women who had caesarean section were discharged <72 hours. Odds of discharge <6 hours among women who gave birth vaginally were lower for births in non-government hospital(aOR = 0.55[95% CI = 0.35;0.85]), and multiple births(aOR = 0.54[95%CI = 0.31;0.94]); while the odds were higher in five of the 8 regions compared to Boké. Among women who gave birth by caesarean section, odds of discharge <72 hours were lower for births in government hospitals (aOR = 0.09[95%CI = 0.03;0.3]), and girl newborns(aOR = 0.15[95%CI = 0.05;0.48]).This study showed that postpartum length-of-stays in Guinea is on average shorter than the local recommendations, with the majority of postpartum women with vaginal births spending less than 6-hours in health facilities after birth. Early discharge was associated with type of facility

website. Authors do not hold ownership of the data and this reason prevents us from sharing the data with the manuscript. However, all DHS data can be accessed after request from the DHS through this link: https://dhsprogram.com/data/dataset_admin/login_main.cfm?CFID=265801198&CFTOKEN=53ab01f71d023a0b-32A9ACCB-F51C-F82D-FBC1F37514D7C836.

**Funding:** This work did not receive any direct funding. The Research Foundation—Flanders (FWO) partly supported LB as part of her Senior Postdoctoral Fellowship. The funders had no role in study design, data collection and analysis, decision to publish, or preparation of the manuscript.

**Competing interests:** The authors have declared that no competing interests exist.

of birth and region. This warrants an in-depth exploration of reasons related to women's and families' preferences, health workers' practices, resource availability, and whether/how early discharge affects postpartum quality-of-care and health outcomes.

## Introduction

The immediate postpartum period, defined as the first 24 hours after birth, represents a critical time for women and newborns. This period involves increased risk of maternal and neonatal morbidity [1] and mortality. Additionally, this period represents a time with several needs for support and guidance to ease the transition from pregnancy to the extended postnatal period and motherhood [2]. Care provided to women and newborns during this period holds potential benefits for their long-term health and wellbeing [3, 4]. Currently the World Health Organization (WHO) recommends a minimum in-facility stay of 24 hours after an uncomplicated vaginal birth in a health facility. The purpose of this stay is to monitor the health of mother and newborn, identify and treat complications that would otherwise go undetected, and subsequently reduce the risk of maternal and perinatal morbidity and mortality. In addition to monitoring, the benefits of staying in the facility after birth can be expanded by providing counselling on breastfeeding, nutrition, family planning and psycho-social wellbeing of the mother, and discharge preparedness [5, 6]. According to the WHO, positive postpartum care extends beyond the first 24 hours to include a minimum of three postnatal contacts within six weeks after discharge [6]. The WHO advises adapting their recommendations into context-appropriate guidelines to meet specific contextual needs at national and regional levels [6].

These global recommendations, along with two Cochrane systematic reviews, acknowledge the lack of sufficient evidence informing the organisation of models of postnatal care (PNC) [7–9]. Models of PNC refer to the organisation of care provision elements, such as length-of-stay after facility-based childbirth, and the location, frequency, timing, duration and intensity of postnatal checks. The ways in which PNC models are conceptualised and organized determines the extent to which their implementation is feasible in various contexts [10]. Barriers to implementing those models, including the lack of women-centred approaches to care, could lead to reduced accessibility and use of PNC services [10]. Therefore, the way in which PNC models are organized and implemented globally is crucial to the health and wellbeing of women and newborns.

Models of PNC care provision, including duration of facility length-of-stay, vary globally and across low- and middle-income countries [11, 12]. Various factors, including clinical obstetric and paediatric risk factors, influence postnatal length-of-stay after facility birth (e.g. postpartum complications, multiplicity of births, newborn prematurity and birth weight, etc.) [12–22]. According to Campbell et al's conceptual framework of determinants of LoS, the availability of skilled human resources (i.e. staffing levels and capacities) and their working conditions, the availability of equipment and supplies in sufficient quantities to support the volume of patients, and space and bed availability in high-volume facilities play a crucial role in the provision, timing and quality of immediate PNC, and in determining length-of-stay after childbirth [12]. Facility-level norms, protocols and guidelines also affect postnatal length-of-stay, and could influence healthcare providers' perceptions and attitudes, which in turn influence practice and adherence to guidelines [12]. Other structural factors include community norms and women's preferences regarding PNC and length-of-stay [12]. However, evidence on structural, community-level and individual determinants of postpartum length-of-stay is lacking in many settings.

Guinea is an example of a country with lack of evidence on immediate PNC, length-of-stay and factors influencing discharge from facilities. The country carries a high burden of maternal morbidity and mortality, with an estimated maternal mortality ratio of 553 deaths per 100,000 live births in 2020 [23]; 34% of which occur in the postnatal period [24]. Almost half of births occurred in a healthcare facility in 2018 (53%), a significant increase from 29% in 1999 [25]. A recent analysis of 2018 DHS data showed that only 20% of women reported receiving all three services along the continuum of maternity care (four ANC visits, a facility-based birth, and a post-partum check within 48h hours) [26]. The population-level caesarean section rate is estimated at a very low level of 2.7%, with notable variability by administrative region, urban/rural residence and socio-economic status [25].

The Guinean Ministry of Health and Public Hygiene's guideline recommends a 24-hour facility stay following childbirth, highlighting the need for frequent monitoring of women's health for at least two hours, and preferably for 6 hours. For home births, the Ministry recommends a consultation with a skilled healthcare provider within the first 24 hours after birth, either at home or in a health facility. Essential services recommended during the first six hours after birth include assessment of vaginal bleeding, checking of maternal blood pressure, temperature, pulse rate and urine void. The guideline recommends early initiation of breastfeeding during the first hour after birth and skin-to-skin contact between mother and newborn [27].

Some barriers to using outpatient PNC were documented in Guinea, and they include financial barriers (e.g., unaffordability of transportation to reach the health facility) [28], health-system related barriers (e.g., shortage of skilled healthcare providers) [28], and lack of communication about the importance of PNC checks [26, 28]. However, there is a lack of evidence about postpartum length-of-stay after facility births and factors associated with discharge time in Guinea.

## Study objectives

This study aims to contribute to the knowledge available on the length-of-stay element of the PNC model available in Guinea. The primary objective is to describe postpartum length-of-stay for women who give birth in a health facility in Guinea in the five-year period (2013–2018), and investigate factors associated with early discharge using Demographic and Health Survey (DHS) data. More specifically, we aim to: A) describe length-of-stay and percentage of early discharge by mode of birth, using international and locally-relevant cut-offs, B) document the proportion of women receiving a health check by a skilled provider during postpartum stays in health facilities; and C) assess characteristics and factors associated with early discharge by mode of birth.

## Methods

### Study setting

Guinea is a lower-middle income country in West Africa, with eight administrative regions, and an estimated population of 13,800,000 in 2022 [29]. The national healthcare system has a pyramidal structure, with the main contribution by the public sector [30]. Childbirth care is offered at all facility levels (primary, secondary and tertiary), and routine out-patient PNC is mostly available at the primary level by midwives and technical health officers [28]. Free emergency obstetric care was introduced in all public health facilities in 2010, including for antenatal care (ANC), vaginal birth and caesarean section, which contributed to a significant decrease in unmet obstetric need, such as documented in Kissidougou health district [31]. However, Guinea's health system was negatively impacted by the 2014–16 Ebola-virus disease epidemic leading to a decline in ANC attendance and facility-based births [32].

## Study design, data, population and sample

This is a quantitative cross-sectional study using publicly available secondary data (S1 STROBE Checklist). We used the most recently published Guinean DHS collected in 2018 [33]. DHS household survey samples are designed to provide nationally-representative estimates. The analysis includes women aged 15–49 years (at the time of the survey) who had their most recent live birth (i.e. excluding stillbirths) in a health facility in the five years preceding the survey (2013–2018). We excluded women whose most recent livebirth was outside of a health facility (n = 2,649), those who had missing/don't know answers on the outcome of interest (length-of-stay, n = 99), and those whose responses were outliers (n = 19). The final sample size was n = 2,763 women (Fig 1).

## Outcome and independent variables

S1 Table presents detailed definitions of variables. The outcome is length-of-stay which represents time that women spent in a health facility after their most recent livebirth and before discharge (i.e. not considering any re-admissions). The DHS collected the variable by asking: *«Combien de temps après l'accouchement de (NOM) êtes-vous restée là? » ["How long after the birth of [name] did you stay here?" ("here"* referring to the facility where birth took place)]. This was collected as a continuous variable, expressed by hours, days and weeks. Values were recoded to generate an average value that accounts for the reporting bias introduced by the unit of time according to Campbell et al. [12], and excludes outliers (extreme values) with length-of-stay>3 weeks (Fig 1, n = 19 values with examples of 48 days, 30 days, 24 weeks, 36 weeks, etc.). The continuous variable was categorised to reflect early discharge depending on women's reported mode of birth. For women who had a vaginal birth, early discharge was defined according to two cut-offs: the first based on the WHO global recommendation and the Guinean official guideline, meaning that a woman was considered to have been discharged early if she reported leaving the facility in less than 24 hours [6]; the second definition was based on local practices and norms, meaning that a woman was considered to have early discharge if she left the facility in less than 6 hours [27]. This locally-informed cut-off reflects an older version of the Guinean guideline which continues to be implemented in most facilities. For women who gave birth by caesarean section, early discharge was defined as leaving the facility in less than 72 hours after birth; this is both an international and Guinean recommendation [6, 27]. We additionally explored additional categories of length-of-stay to describe with nuance the distribution of length-of-stay among those who stayed "too short" (women who had vaginal birth: <2 hours; 2–3 hours; 4–23 hours; ≥24 hours) and those who stayed "too long" (women who had caesarean section: <24 hours; 24–71 hours; 72–167 hours; ≥168 hours). We additionally explored whether women reported receiving a postnatal health check at the facility by a skilled care provider. Women were considered to have received a check if the provider was a doctor, nurse-midwife or technical health officer (S1 Table).

Independent variables were identified following a thorough literature review and relying on Campbell et al's conceptual framework on postpartum length-of-stay [12]. We included factors theoretically linked to postpartum length-of-stay and available in the DHS, and span five levels: 1-Community & family factors (region, place of residence, ethnicity, marital and cohabiting status, involvement in decision-making regarding own healthcare, number of household members, relationship to head of household); 2-Facility characteristics & norms (type of facility, skilled attendance at birth, birth on weekday/weekend; 3-Women's socio-economic characteristics (maternal age at birth, education level, occupation at time of survey, household-level wealth index, health insurance and phone ownership, issues perceived as big problems to access healthcare); 4-Women's needs and obstetric history (parity, antenatal care frequency

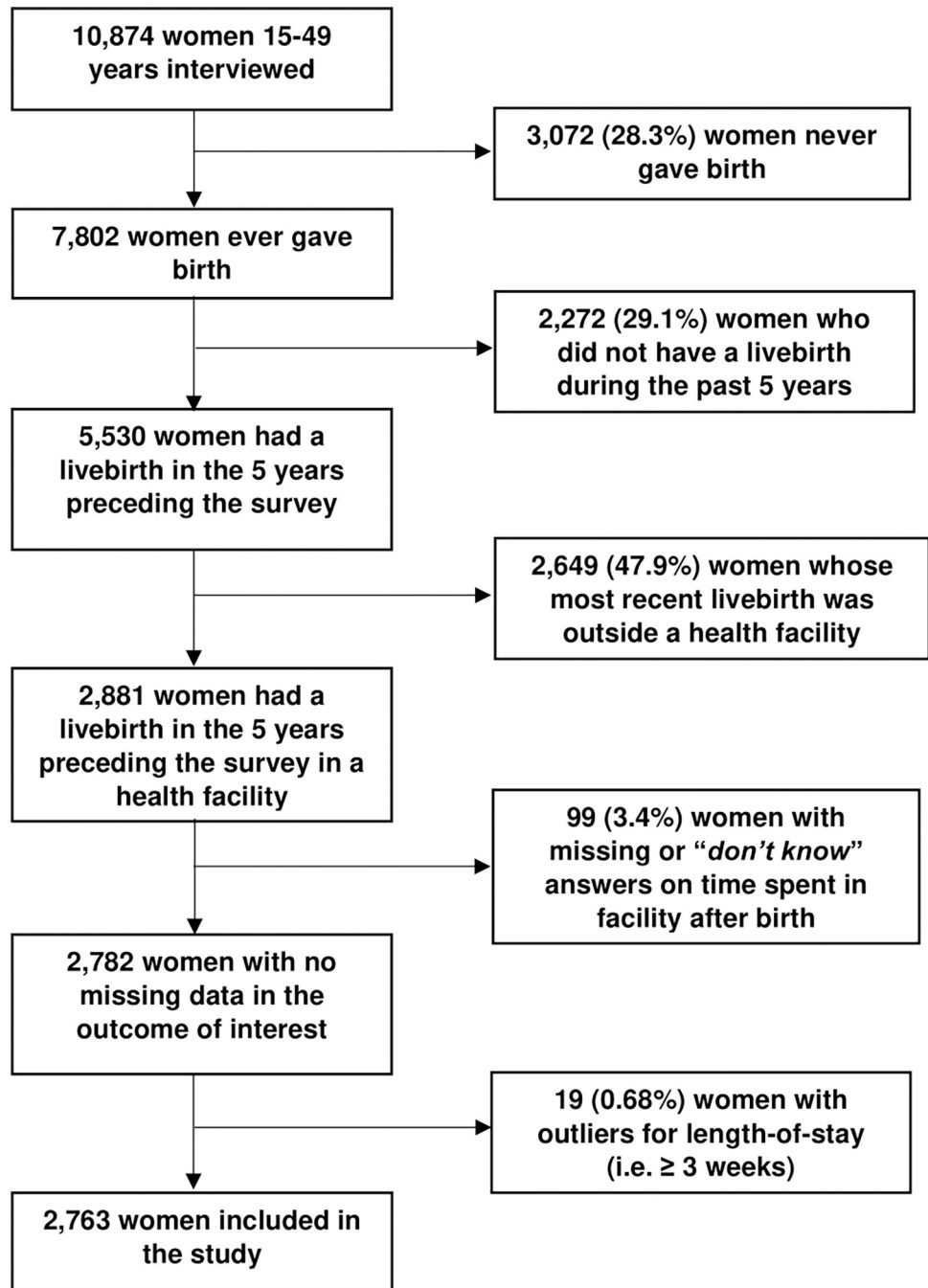

**Fig 1. Flowchart of inclusion criteria and final sample size–Guinea DHS 2018 (unweighted percentages shown).**

and timing of first visit, multiple births, whether the pregnancy was wanted, history of previous miscarriage, abortion or stillbirth); and 5-Newborn characteristics (sex, perceived size at birth, newborn survival and time of death). Facility type was defined using a recategorization of the DHS categories to reflect government hospitals (including regional and prefectorial); government lower-level facilities (including health posts, health centres, and other public sector facilities); non-governmental hospital (private hospital); and non-governmental lower-level facility (clinic AGBEF, private care clinic).

## Analysis

We described sample characteristics by mode of birth, and summarized frequencies and percentages with 95% confidence intervals (CI) for categorical variables. Length-of-stay was described separately by mode of birth (vaginal birth and birth by caesarean section), both as a continuous variable in hours (mean and median), and categorical as early or not (frequencies and percentages). We displayed the percentage of early discharge after vaginal birth in each of the 8 administrative regions on 2 maps (1 for each <24 hours and <6 hours). We were not able to conduct this analysis for caesarean sections due to small sample size. Results were displayed in tables and visualised in graphs.

Associations between early discharge and independent variables were explored with simple logistic regression. Factors with p-value<0.2 were included in the multi-variable logistic regression models, separately by mode of birth. The model for early discharge among women who gave birth by caesarean birth was limited to five factors selected based on theoretical grounds because of sample size limitations. We checked for models' assumptions including model specification and variance inflation factor. Data analysis was conducted in Stata SE v.16. Statistical significance level was set at p-value<0.05. All analyses adjusted for the stratified sampling strategy and sampling weights of the DHS. A sensitivity analysis explored including women who had outlier values for length-of-stay and showed no differences in the distribution of sample characteristics and outcome (S2 Table).

## Missing data

Ninety-nine women with missing length-of-stay were excluded from the analysis (Fig 1), among whom 28 had missing length-of-stay because the mode of birth variable was missing (system-missing), and 71 answered "don't know" to the length-of-stay question. We described the socio-demographic characteristics for these women in comparison to those where data was available in Table 2.1 in S2 Table. The results show that differential missingness exists to a certain extent in terms of region (majority from Mamou and Nzérékoré, predominantly rural regions), residence (majority in rural settings), type of facility where birth took place, education, occupation, mobile phone ownership, and higher parity. Two independent variables had missing values; n = 276 missing values on involvement in decision-making regarding own healthcare, and n = 19 missing values on perceived newborn size at birth. Among these two variables, only the perceived newborn size at birth was eligible to be included in the multi-variable logistic regression model on early discharge among vaginal births. We conducted a sensitivity analysis by excluding the variable from the model (S3 Table), which showed no major changes in findings.

## Ethics

This secondary data analysis was waved from an ethics review by the ethics committee at the Institute of Tropical Medicine, Antwerp. DHS protocols and questionnaires were reviewed and approved by ICF institutional review board and by a review board in Guinea. Prior to participation, all participants provided informed consent emphasizing voluntary participation, and parents/guardians of adolescent participants provided consent. The conduct of data collection and processing ensures protection of privacy and confidentiality of participants [34]. The analysed secondary data were anonymised to protect participant confidentiality.

## Results

### Sample characteristics

Table 1 describes characteristics of 2,763 women who had their most recent livebirth in the five years preceding the survey in a health facility, by mode of birth. The majority—2,603 women (94.5% [95%CI = 93.5;95.4]) reported giving birth vaginally, and 160 (5.5% [95% CI = 4.5;6.5]) gave birth via caesarean section. About one-fifth resided in Conakry and Nzérékoré each, and almost half resided in urban settings. Most women were living with a partner at the time of the survey, and 42.3% [95%CI = 38.7; 46.0] reported being partially or completely involved in decision-making regarding own healthcare. Most births occurred in a government lower-level facility and were attended by a skilled provider. Average maternal age at birth was 27 years (se = 0.15 years), and two-thirds of women reported having no education 65.8% [95% CI = 63.2; 68.3]. About two-fifths of women had a parity of 4 or more, and almost half attended at least 4 ANC visits. The index birth was a multiple birth among 2.8% [95%CI = 2.2; 3.6] of women.

### Description of the outcome: Postpartum length-of-stay

Table 2 describes postpartum length-of-stay. The mean time spent in the facility among women with a vaginal birth was 8.5 hours (se = 0.47), and the median was 3 hours (IQR = [2; 4]). The majority (81.5% [95%CI = 79.6; 83.2]) were discharged in less than 6 hours; almost half spent 2–3 hours in the facility.

The mean time spent in the facility among women who gave birth via caesarean section was 5.5 days (131.7 hours, se = 10.5). Early discharge (<72 hours) was noted among 28% [95% CI = 19.3; 38.7]. Most women who gave birth by caesarean section spent more than 24 hours in the facility (78.8% [95%CI = 67.8; 86.8]), and 32.2% [95%CI = 24.7; 40.8] spent more than seven days (i.e. 168 hours) in the facility.

Fig 2 shows maps with the percentage of women who were discharged early after vaginal birth according to the WHO recommendation of 24 hours (Fig 2A) and the local recommendation of 6 hours (Fig 2B), in each of the 8 administrative regions in Guinea. The percentage of early discharge before 24 hours ranged between 78.2% [95%CI = 71.3; 83.9] of women in Labé and 95.4% [95%CO = 93.0; 97.0] in Kankan. The percentage of early discharge <6 hours ranged between 63.3% [95%CI = 55.6; 70.3] in Labé and 89.2% [95% CI = 86.0; 91.7] in Nzérékoré. The smallest discrepancy between the two cut-offs was in Nzérékoré (91.1% [95%CI = 87.9; 93.5] early discharge <24 hours vs. 89.2% [95%CI = 86.0; 91.7] early discharge <6 hours). The largest discrepancy between the two cut-offs was in Boké with 82.7% [95%CI = 76.1; 87.7] of women discharged early according to WHO recommendations, vs. 64.7% [95%CI = 55.6; 72.9] according to local recommendations.

Fig 3 shows the percentage of women who reported receiving a health check by a skilled provider before discharge from the facility, by mode of birth and time of discharge. 7.8% [95% CI = 3.9; 14.6] of women who had caesarean section and 25.6% [95%CI = 22.7; 28.7] of women who had a vaginal birth reported not receiving a health check before discharge. About one-third (32.4% [95%CI = 27.6; 37.6]) of women who had a vaginal birth and spent less than 2 hours in the facility reported not receiving a health check before discharge. This percentage decreased as length-of-stay increased, to reach 21.3% [95%CI = 16.3; 27.3] among those who spent ≥24 hours in the facility. Among women who gave birth by caesarean section and were discharge before 24 hours, 17.6% [95%CI = 4.3; 36.4] reported not receiving a health check by a skilled provider.

**Table 1. Characteristics of women who gave their most recent livebirth in a health facility in the five years preceding the Guinea DHS 2018, by mode of birth (n = 2,763).**

| | | Vaginal birth (n = 2,603) | | Caesarean section birth (n = 160) | | Total (n = 2,763) | |
|---|---|---|---|---|---|---|---|
| | Characteristics | n | % [95%CI] | n | % [95%CI] | n | % [95%CI] |
| **Community and family factors** | **Region** | | | | | | |
| | Boké | 321 | 8.8 [7.1; 10.9] | 22 | 11.1 [7.9; 15.3] | 343 | 8.9 [7.2; 11.0] |
| | Conakry | 420 | 20.1 [17.9; 22.5] | 33 | 25.5 [21.2; 30.3] | 453 | 20.4 [18.3; 22.7] |
| | Faranah | 236 | 6.7 [5.5; 8.1] | 14 | 6.5 [4.8; 8.7] | 250 | 6.7 [5.5; 8.0] |
| | Kankan | 417 | 17.2 [14.3; 20.5] | 13 | 9.2 [6.9; 12.3] | 430 | 16.7 [13.9; 20.0] |
| | Kindia | 372 | 14.9 [12.7; 17.5] | 18 | 13.1 [10.1; 16.6] | 390 | 14.9 [12.6; 17.4] |
| | Labé | 219 | 6.7 [5.5; 17.5] | 18 | 9.2 [6.2; 13.4] | 237 | 6.9 [5.6; 8.3] |
| | Mamou | 191 | 5.2 [4.1; 17.5] | 16 | 5.9 [4.3; 8.0] | 207 | 5.2 [4.1; 6.6] |
| | Nzérékoré | 427 | 20.4 [17.4; 17.5] | 26 | 19.5 [14.1; 26.4] | 453 | 20.3 [17.4; 23.6] |
| | **Residence** | | | | | | |
| | Rural | 1364 | 53.2 [47.6; 58.7] | 56 | 35.5 [26.3; 45.8] | 1420 | 52.2 [46.6; 57.7] |
| | Urban | 1239 | 49.8 [41.3; 52.4] | 104 | 64.5 [54.2; 73.7] | 1343 | 47.8 [42.3; 53.3] |
| | **Ethnicity** | | | | | | |
| | Soussou | 571 | 22.2 [19.2; 25.6] | 29 | 19.6 [13.5; 26.9] | 600 | 22.1 [19.3; 25.3] |
| | Peuls | 812 | 26.5 [23.6; 29.6] | 59 | 32.6 [25.5; 40.7] | 871 | 26.8 [23.9; 29.9] |
| | Malinké | 825 | 32.2 [28.6; 36.2] | 46 | 29.8 [23.3; 37.2] | 871 | 32.1 [28.5; 35.9] |
| | Other (Kissi, Toma, Guerzé) | 395 | 19.1 [15.4; 23.2] | 26 | 17.9 [11.9; 26.2] | 421 | 19.0 [15.5; 23.1] |
| | **Marital and cohabiting status** | | | | | | |
| | Not in union/not living with a partner | 613 | 23.2 [21.1; 25.3] | 42 | 26.3 [19.4; 34.5] | 655 | 23.3 [21.3; 25.4] |
| | Living with a partner | 1990 | 76.8 [74.7; 78.9] | 118 | 73.7 [65.5; 80.6] | 2108 | 76.7 [74.6; 78.7] |
| | **Involved in DM re own healthcare*** | 936 | 42.4 [38.8; 45.9] | 59 | 41.7 [32.9; 50.9] | 995 | 42.3 [38.7; 46.0] |
| | **Number of household members** | | | | | | |
| | 2–3 members | 211 | 8.2 [7.2; 9.4] | 12 | 6.7 [3.6; 12.3] | 223 | 8.1 [7.1; 9.3] |
| | 4–9 members | 1737 | 67.3 [64.6; 96.8] | 105 | 66.3 [57.4; 74.1] | 1842 | 67.2 [64.6; 69.7] |
| | 10 or more members | 655 | 24.5 [21.3; 27.3] | 43 | 27.0 [19.7; 35.7] | 698 | 24.7 [22.1; 27.4] |
| | **Relation to head of the household** | | | | | | |
| | Self | 154 | 6.1 [5.2; 7.3] | 13 | 7.5 [4.4; 12.5] | 167 | 6.2 [5.3; 7.3] |
| | Partner | 1784 | 69.0 [66.6; 71.1] | 109 | 69.8 [61.6; 76.9] | 1893 | 68.9 [66.7; 71.1] |
| | Child/child in law | 446 | 17.0 [15.3; 18.8] | 23 | 13.4 [8.7; 20.0] | 469 | 16.8 [15.2; 18.6] |
| | Other | 219 | 7.9 [6.8; 9.2] | 15 | 9.3 [5.5; 15.2] | 234 | 8.1 [6.9; 9.3] |
| **Facility characteristics and norms** | **Type of facility** | | | | | | |
| | Government lower level facility | 1818 | 71.8 [68.4; 74.8] | 52 | 31.9 [23.1; 42.3] | 1870 | 69.6 [66.3; 72.7] |
| | Government hospital | 513 | 17.9 [15.6; 20.7] | 93 | 58.1 [48.1; 67.4] | 605 | 20.1 [17.6; 22.9] |
| | Non-government lower level facility | 71 | 2.5 [1.8; 3.5] | 3 | 1.5 [0.5; 4.5] | 74 | 2.5 [1.8; 3.4] |
| | Non-government hospital | 201 | 7.8 [6.4; 9.3] | 13 | 8.5 [4.9; 14.5] | 214 | 7.8 [6.5; 9.3] |
| | **Skilled attendance at birth** | 2432 | 94.8 [92.2; 96.6] | 155 | 94.7 [85.6; 98.2] | 2647 | 94.8 [92.2; 96.6] |
| | **Day of birth** | | | | | | |
| | Weekday | 1808 | 69.0 [67.1; 70.9] | 117 | 72.4 [64.3; 79.2] | 1925 | 69.2 [67.3; 71.0] |
| | Weekend | 795 | 31.0 [29.1; 32.9] | 43 | 27.6 [20.8; 35.7] | 838 | 30.8 [29.0; 32.7] |

*(Continued)*

**Table 1.** (*Continued*)

| | Characteristics | Vaginal birth (n = 2,603) | | Caesarean section birth (n = 160) | | Total (n = 2,763) | |
|---|---|---|---|---|---|---|---|
| | | n | % [95%CI] | n | % [95%CI] | n | % [95%CI] |
| **Women's socio-economic characteristics** | **Maternal age at birth** (mean, se) | 26.96 | 0.15 | 28.2 | 0.7 | 27.0 | 0.15 |
| | **Maternal age at birth** (in years) | | | | | | |
| | 13–19 years | 422 | 16.3 [14.9; 17.7] | 22 | 12.5 [7.9; 18.9] | 444 | 16.1 [14.7; 17.5] |
| | 20–24 years | 582 | 22.7 [20.9; 24.6] | 23 | 13.8 [8.7; 21.1] | 605 | 22.2 [20.5; 24.0] |
| | 25–29 years | 641 | 24.6 [23.0; 26.3] | 49 | 31.5 [24.6; 39.3] | 690 | 24.9 [23.4; 26.6] |
| | 30–34 years | 471 | 17.9 [16.4; 19.5] | 32 | 19.8 [13.7; 27.7] | 503 | 18.0 [16.5; 19.5] |
| | 35–49 years | 487 | 18.6 [16.9; 20.3] | 34 | 22.4 [15.5; 31.3] | 521 | 18.8 [17.2; 20.4] |
| | **Highest education level reached** | | | | | | |
| | No education | 1715 | 66.5 [63.9; 69.1] | 87 | 53.2 [44.7; 61.6] | 1802 | 65.8 [63.2; 68.3] |
| | Primary education | 355 | 13.0 [11.5; 14.6] | 25 | 15.3 [10.1; 22.7] | 380 | 13.1 [11.6; 14.7] |
| | Secondary or higher | 533 | 20.5 [18.5; 22.7] | 48 | 31.5 [24.6; 39.2] | 581 | 21.1 [19.1; 23.3] |
| | **Occupation frequency at time of survey** | | | | | | |
| | Not worked in the past 12 months | 683 | 24.7 [22.3; 27.1] | 47 | 27.5 [20.3; 35.9] | 730 | 24.8 [22.5; 27.3] |
| | Occasional | 338 | 12.7 [11.1; 14.5] | 11 | 7.5 [4.1; 13.3] | 349 | 12.4 [10.8; 14.2] |
| | Seasonal | 513 | 21.0 [18.2; 24.1] | 17 | 12.0 [7.3; 19.2] | 530 | 20.6 [17.9; 23.5] |
| | All year | 1069 | 41.6 [38.6; 44.6] | 85 | 53.1 [44.2; 61.7] | 1154 | 42.2 [39.3; 45.2] |
| | **Household wealth index** | | | | | | |
| | Poorest | 454 | 16.7 [13.7; 20.3] | 27 | 16.6 [10.9; 24.6] | 481 | 16.7 [13.8; 20.1] |
| | Poorer | 517 | 19.2 [17.2; 21.4] | 22 | 11.9 [7.7; 17.7] | 539 | 18.8 [16.8; 20.9] |
| | Middle | 521 | 20.8 [18.8; 23] | 29 | 19.9 [13.5; 28.2] | 550 | 20.8 [18.7; 22.9] |
| | Richer | 535 | 21.3 [18.9; 23.7] | 32 | 20.4 [14.2; 28.5] | 567 | 21.2 [19; 23.6] |
| | Richest | 576 | 22.0 [18.8; 25.5] | 50 | 31.2 [22.6; 41.4] | 626 | 22.5 [19.3; 25.9] |
| | **Owns health insurance** | 46 | 1.8 [1.2; 2.7] | 6 | 4.3 [1.7; 10.3] | 52 | 2.0 [1.3; 2.97] |
| | **Owns mobile phone** | 2036 | 77.1 [74.2; 79.7] | 146 | 91.6 [85.7; 95.2] | 2182 | 77.9 [75.1; 80.4] |
| | **Issue perceived as a big problem to seek care for self** | | | | | | |
| | Distance to health facility | 934 | 35.2 [32.2; 38.4] | 42 | 23.9 [17.2; 32.1] | 976 | 34.6 [31.7; 37.7] |
| | Getting permission to go | 591 | 21.4 [18.8; 24.3] | 34 | 21.0 [14.8; 28.9] | 625 | 21.4 [18.8; 24.2] |
| | Getting money needed for treatment | 1451 | 55.4 [52.3; 58.4] | 71 | 37.7 [29.9; 46.1] | 1522 | 54.4 [51.4; 57.4] |
| | Not wanting to go alone | 610 | 22.7 [20.3; 25.4] | 35 | 19.9 [13.9; 27.9] | 645 | 22.6 [20.2; 25.2] |
| **Women's needs and obstetrics history** | **Parity at index birth** | | | | | | |
| | Primiparous | 616 | 23.9 [22.2; 25.6] | 47 | 27.4 [20.5; 35.7] | 663 | 24.1 [22.4; 25.8] |
| | Multiparous 2–3 | 969 | 36.7 [34.6; 38.9] | 60 | 39.4 [30.7; 46.6] | 1029 | 36.8 [34.8; 38.9] |
| | Multiparous 4 or more | 1018 | 39.4 [37.3; 41.5] | 53 | 34.2 [26.1; 43.3] | 1071 | 39.1 [37.1; 41.2] |
| | **ANC visits during pregnancy** | | | | | | |
| | None | 84 | 2.9 [2.1; 3.8] | 4 | 3.3 [1.0; 10.1] | 88 | 2.9 [2.2; 3.8] |
| | 1–3 visits | 1318 | 50.4 [47.8; 53] | 64 | 39.7 [31.6; 48.3] | 1382 | 49.8 [47.3; 52.3] |
| | 4 or more visits | 1201 | 46.7 [44.1; 49.4] | 92 | 57.0 [48.2; 65.5] | 1293 | 47.3 [44.6; 49.9] |
| | **Timing of first ANC visit** | | | | | | |
| | None | 84 | 2.9 [2.1; 3.8] | 4 | 3.3 [1; 10.1] | 88 | 2.9 [2.2; 3.8] |
| | During 1st trimester | 899 | 34.3 [31.6; 37.2] | 54 | 28.8 [21.9; 36.9] | 953 | 34.0 [31.3; 36.8] |
| | Beyond 1st trimester | 1620 | 62.8 [59.8; 65.7] | 102 | 67.9 [59.6; 75.2] | 1722 | 63.1 [60.1; 65.9] |
| | **Multiple birth** | 74 | 2.8 [2.2; 3.6] | 6 | 3.2 [1.4; 7.1] | 80 | 2.8 [2.2; 3.6] |
| | **Pregnancy wanted at the time** | 2188 | 84.2 [82.6; 85.8] | 136 | 84.7 [77.4; 90] | 2324 | 84.3 [82.6; 85.8] |
| | **Ever had a terminated pregnancy** | 274 | 10.1 [8.8; 11.6] | 33 | 21.3 [14.9; 29.6] | 307 | 10.7 [9.4; 12.2] |

(*Continued*)

**Table 1.** (Continued)

| | Characteristics | Vaginal birth (n = 2,603) | | Caesarean section birth (n = 160) | | Total (n = 2,763) | |
|---|---|---|---|---|---|---|---|
| | | n | % [95%CI] | n | % [95%CI] | n | % [95%CI] |
| Newborn characteristics | Newborn sex | | | | | | |
| | Girl | 1258 | 48.3 [46.3; 50.2] | 85 | 52.7 [44.3; 60.9] | 1343 | 48.5 [46.6; 50.4] |
| | Boy | 1345 | 51.7 [49.8; 53.7] | 75 | 47.3 [39; 55.7] | 1420 | 51.5 [49.2; 53.4] |
| | Perceived size at birth** | | | | | | |
| | Smaller than average | 210 | 7.9 [6.7; 9.2] | 15 | 11.2 [6.5; 18.6] | 225 | 8.1 [6.9; 9.4] |
| | Average or larger | 2375 | 92.1 [90.8; 93.3] | 144 | 88.8 [81.4; 93.5] | 2519 | 91.9 [90.6; 93.1] |
| | Newborn survival | | | | | | |
| | Survived until survey | 2479 | 95.1 [94; 96.1] | 151 | 94.0 [87.9; 97.1] | 2630 | 95.0 [93.9; 96] |
| | Died on/before discharge | 21 | 0.7 [0.4; 1.1] | 4 | 3.9 [1.4; 10.4] | 25 | 0.9 [0.6; 1.3] |
| | Died after discharge | 103 | 4.2 [3.3; 5.3] | 5 | 2.1 [0.8; 5.6] | 108 | 4.1 [3.2; 5.1] |

*276 missing values-partially or completely involved

**19 missing values

## Factors associated with early discharge: Binary and multivariable logistic regression

### 1. Women who gave a vaginal birth

Table 3 shows the results of the binary and multivariable logistic regression assessing factors associated with discharge <6hours among women who had their most recent livebirth

**Table 2. Postpartum length-of-stay (both continuous and categorical) among women who gave their most recent livebirth in a health facility in the five years preceding the Guinea DHS 2018, by mode of birth (n = 2,763).**

| | Vaginal birth n = 2603 | | Caesarean section n = 160 | | Total n = 2763 | |
|---|---|---|---|---|---|---|
| Length-of-stay (continuous) in hours (mean, se) | 8.5 | se = 0.47 | 131.7 | se = 10.5 | 15.2 | se = 0.88 |
| Length-of-stay (continuous) in hours (median, IQR) | 3 | IQR = [2; 4] | 108 | IQR = [60; 204] | 3 | IQR = [2; 5] |
| | n | % [95%CI] | n | % [95%CI] | n | % [95%CI] |
| **Length-of-stay (categorical)** | | | | | | |
| **<24 hours** | **2321** | **90.0 [88.7; 91.2]** | **31** | **21.2 [13.2; 32.2]** | **2352** | **86.3 [84.8; 87.6]** |
| <2 hours | 534 | 20.3 [18.3; 22.6] | | | 539 | 19.5 [17.4; 21.5] |
| 2–3 hours | 1165 | 47.0 [44.4; 49.7] | | | 1183 | 45.2 [42.3; 47.5] |
| 4–23 hours | 622 | 22.6 [20.7; 24.7] | | | 630 | 21.6 [12.0; 14.9] |
| **≥24 hours** | **282** | **10.0 [8.8; 11.3]** | **129** | **78.8 [67.8; 86.8]** | **411** | **13.7 [12.4; 15.2]** |
| 24–71 hours | | | 10 | 6.7 [3.4; 12.9] | 247 | 8.4 [7.2; 9.6] |
| 72–167 hours | | | 66 | 39.8 [31.2; 49.1] | 97 | 3.2 [2.6; 4.0] |
| ≥168 hours | | | 53 | 32.2 [24.7; 40.8] | 67 | 2.1 [1.6; 2.8] |
| **WHO/Guinean guideline cut-off †** | | | | | | |
| Early discharge | 2321 | 90.0 [88.7; 91.2] | 41 | 28.0 [19.3; 38.7] | 2362 | 86.6 [85.2; 87.9] |
| Recommended length-of-stay | 282 | 10.0 [8.8; 11.3] | 119 | 72.0 [63.3; 80.7] | 401 | 13.4 [12.1; 14.8] |
| **Practice informed cut-off ‡** | | | | | | |
| Early discharge | 2072 | 81.5 [79.6; 83.2] | 41 | 28.0 [19.3; 38.7] | 2113 | 78.6 [76.7; 80.4] |
| Recommended length-of-stay | 531 | 18.5 [16.8; 20.4] | 119 | 72.0 [63.3; 80.7] | 650 | 21.4 [19.7; 23.3] |

† Early discharge before 24hrs for vaginal births and before 72hrs for c-section

‡ Early discharge before 6hrs for vaginal births and before 72hrs for c-section

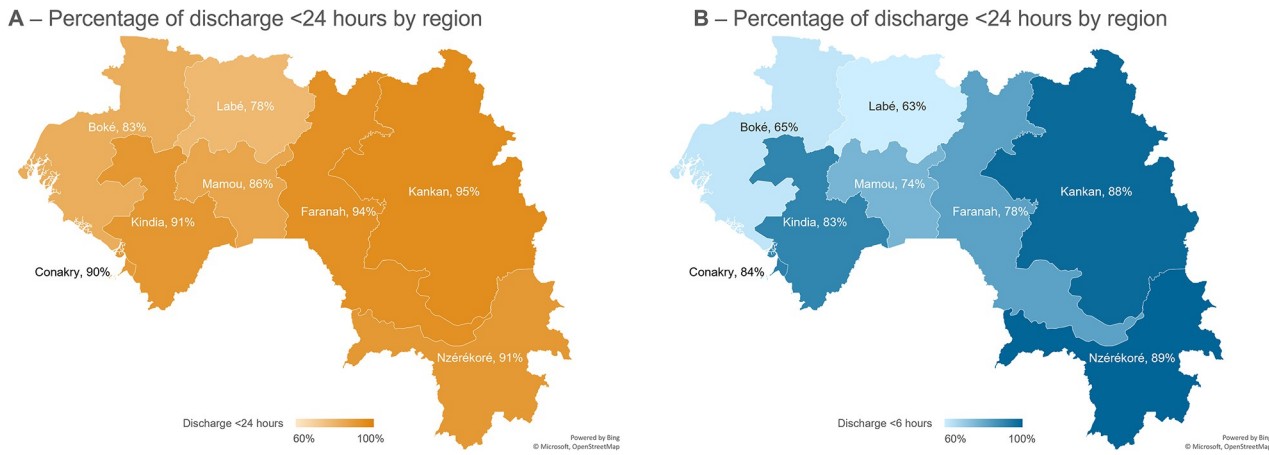

Maps were created with Miscrosoft Excel. The base layers were pulled from Open StreetMap which is open access (CC-BY-SA 2.0) https://www.openstreetmap.org/search?query=guinea#map=7/9.947/-11.602

**Fig 2. Percentage of women with early discharge according to the WHO recommended (24hrs–Fig 2A) and locally relevant cut-offs (6hrs, Fig 2B), among women who gave their most recent vaginal birth in a health facility in the five-years preceding the Guinea DHS2018 (n = 2,603), by region.**

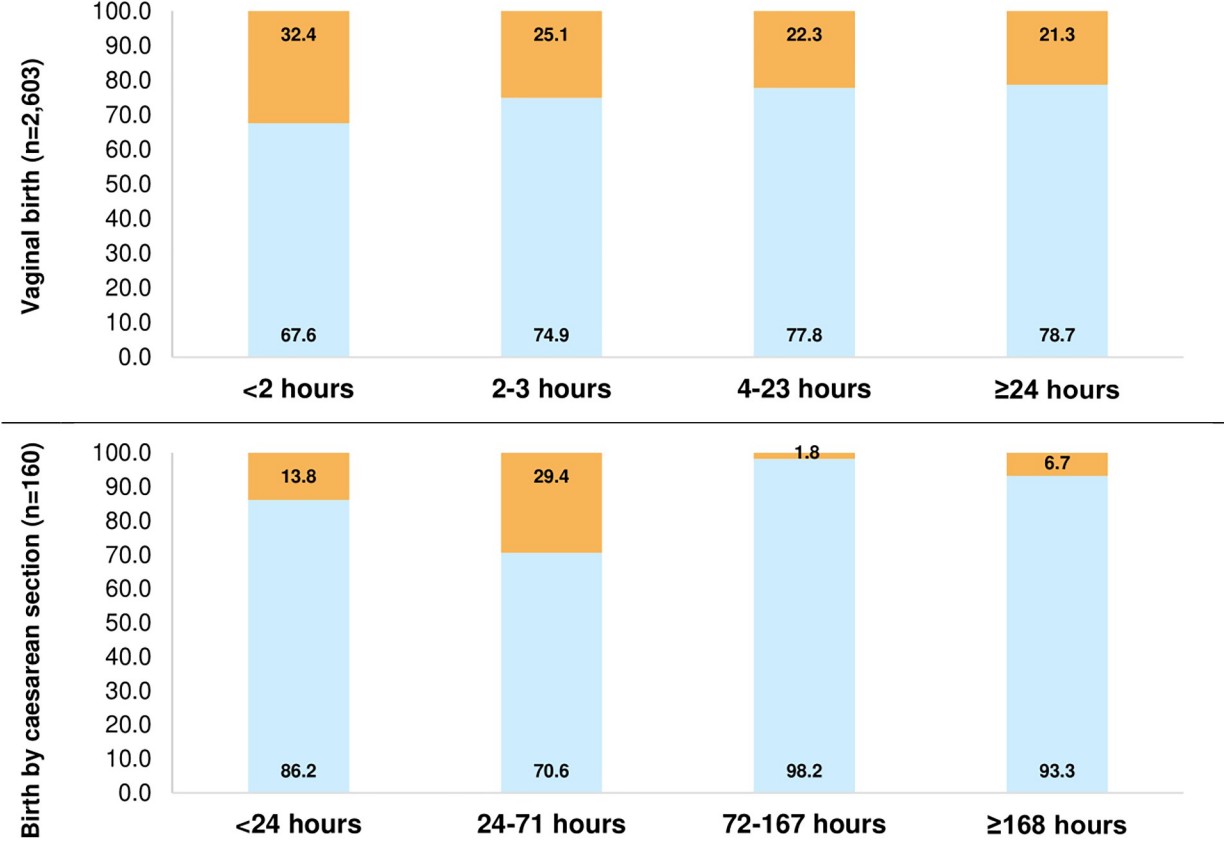

**Fig 3. Percentage of women who reported receiving at least one health check from a skilled provider before discharge, by mode of birth and length-of-stay category, among women who had their most recent livebirth in a health facility, in the five-years preceding the Guinea DHS2018, n = 2,763.**

**Table 3. Binary and multivariable† logistic regression assessing factors associated with early discharge (<6hours, n = 2072) among women who gave their most recent livebirth vaginally in a health facility in the five years preceding the Guinea DHS2018 (total n = 2,603).**

| | Characteristics | Early discharge <6 hours | | | | | |
|---|---|---|---|---|---|---|---|
| | | n (%) | [95%CI] | cOR [95%CI] | p-value | aOR [95%CI] | p-value |
| Community and family factors | **Region** | | | | | | |
| | Boké | 207 (64.7%) | [55.5; 72.9] | ref | | ref | |
| | Conakry | 350 (83.6%) | [78.9; 87.4] | 2.78 [1.67; 4.56] | <0.001 | 3.60 [2.16; 6.01] | <0.001* |
| | Faranah | 186 (77.6%) | [69.8; 83.9] | 1.88 [1.08; 3.29] | 0.026 | 1.98 [1.07; 3.68] | 0.031* |
| | Kankan | 367 (88.1%) | [83.3; 91.6] | 4.02 [2.32; 6.96] | <0.001 | 4.04 [2.1; 7.78] | <0.001* |
| | Kindia | 308 (83.0%) | [78.5; 86.8] | 2.67 [1.64; 4.32] | <0.001 | 2.83 [1.72; 4.65] | <0.001* |
| | Labé | 134 (63.3%) | [55.6; 70.3] | 0.94 [0.57; 1.54] | 0.802 | 1.01 [0.57; 1.79] | 0.984 |
| | Mamou | 139 (74.1%) | [66.8; 80.1] | 1.56 [0.93; 2.61] | 0.094 | 1.71 [0.93; 3.15] | 0.081 |
| | Nzérékoré | 381 (89.2%) | [86.0; 91.7] | 4.49 [2.77; 7.30] | <0.001 | 4.72 [2.39; 9.34] | <0.001* |
| | **Residence** | | | | | | |
| | Rural | 1070 (80.8%) | [78.1; 83.3] | ref | | | |
| | Urban | 1002 (82.3%) | [79.5; 84.8] | 1.1 [0.86; 1.43] | 0.445 | | |
| | **Ethnicity** | | | | | | |
| | Soussou | 438 (80.0%) | [75.6; 83.7] | ref | | ref | |
| | Peuls | 596 (74.6%) | [71.2; 77.8] | 0.74 [0.54; 0.99] | 0.049 | 0.96 [0.66; 1.39] | 0.825 |
| | Malinké | 699 (85.3%) | [82.3; 87.8] | 1.45 [1.04; 2.03] | 0.028 | 0.91 [0.59; 1.38] | 0.650 |
| | Other (Kissi, Toma, Guerzé) | 339 (86.4%) | [82.5; 89.5] | 1.69 [1.07; 2.36] | 0.020 | 0.83 [0.45; 1.52] | 0.456 |
| | **Marital and cohabiting status** | | | | | | |
| | Not in union/not living with a partner | 471 (78.5%) | [74.6; 81.8] | ref | | ref | |
| | Living with a partner | 1601 (82.4%) | [80.4; 84.3] | 1.29 [1.00; 1.64] | 0.042 | 1.08 [0.81; 1.43] | 0.611 |
| | **Involvement in decision making regarding own healthcare** | | | | | | |
| | Not involved | 1119 (81.3%) | [78.9; 83.5] | ref | | | |
| | Involved partly or completely | 754 (83.3%) | [80.4; 85.8] | 1.15 [0.91; 1.46] | 0.243 | | |
| | **Number of household members** | | | | | | |
| | 2–3 members | 178 (85.2%) | [79.4; 89.6] | ref | | | |
| | 4–9 members | 1381 (81.5%) | [79.0; 83.7] | 0.76 [0.49; 1.17] | 0.215 | | |
| | 10 or more members | 513 (80.4%) | [76.9; 83.4] | 0.71 [0.45; 1.12] | 0.140 | | |
| | **Relation to head of the household** | | | | | | |
| | Self | 121 (80.5%) | [73.5; 86.0] | ref | | | |
| | Partner | 1442 (82.5%) | [80.4; 84.5] | 1.14 [0.75; 1.73] | 0.531 | | |
| | Child/child in law | 341 (78.9%) | [74.6; 82.7] | 0.91 [0.57; 1.43] | 0.676 | | |
| | Other | 168 (78.9%) | [72.5; 84.1] | 0.91 [0.53; 1.54] | 0.711 | | |
| Facility characteristics and norms | **Type of facility** | | | | | | |
| | Government lower level facility | 1487 (83.5%) | [81.3; 85.5] | ref | | ref | |
| | Government hospital | 384 (76.6%) | [71.9; 80.6] | 0.64 [0.48; 0.86] | 0.003 | 0.76 [0.56; 1.04] | 0.085 |
| | Non-government lower level facility | 62 (88.6%) | [79.3; 94.0] | 1.53 [0.75; 3.15] | 0.242 | 2.12 [0.95; 4.76] | 0.066 |
| | Non-government hospital | 139 (72.2%) | [65.0; 78.4] | 0.51 [0.35; 0.75] | 0.001 | 0.55 [0.35; 0.85] | 0.008* |
| | **Skilled attendance at birth** | | | | | | |
| | No | 91 (85.8%) | [75.1; 92.4] | ref | | | |
| | Yes | 1981 (81.3%) | [79.3; 83.0] | 0.72 [0.35; 1.46] | 0.360 | | |
| | **Day of birth** | | | | | | |
| | Weekday | 1435 (81.3%) | [79.1; 83.4] | 0.96 [0.78; 1.19] | 0.738 | | |
| | Weekend | 637 (81.9%) | [79.2; 84.3] | ref | | | |

*(Continued)*

**Table 3.** (Continued)

| | Characteristics | | Early discharge <6 hours | | | | |
| --- | --- | --- | --- | --- | --- | --- | --- |
| | Characteristics | n (%) | [95%CI] | cOR [95%CI] | p-value | aOR [95%CI] | p-value |
| **Women's socio-economic characteristics** | **Maternal age at birth (in years)** | | | | | | |
| | 13–19 years | 330 (81.4%) | [77.0; 85.1] | ref | | | |
| | 20–24 years | 471 (81.1%) | [77.3; 84.3] | 0.98 [0.68; 1.40] | 0.905 | | |
| | 25–29 years | 506 (80.7%) | [76.7; 84.1] | 0.95 [0.69; 1.31] | 0.768 | | |
| | 30–34 years | 372 (81.7%) | [77.8; 85.1] | 1.02 [0.72; 1.44] | 0.902 | | |
| | 35–49 years | 393 (83.0%) | [79.2; 86.3] | 1.12 [0.78; 1.60] | 0.543 | | |
| | **Highest education level reached** | | | | | | |
| | No education | 1402(83.5%) | [81.5; 85.4] | ref | | ref | |
| | Primary education | 265 (77.0%) | [72.2; 81.3] | 0.66 [0.50; 0.88] | 0.005 | 0.74 [0.53; 1.02] | 0.065 |
| | Secondary or higher | 405 (77.8%) | [73.5; 81.6] | 0.69 [0.53; 0.90] | 0.006 | 0.76 [0.57; 1.02] | 0.065 |
| | **Occupation frequency at time of survey** | | | | | | |
| | Not worked in the past 12 months | 522 (78.7%) | [75.0; 81.9] | ref | | ref | |
| | Occasional | 263 (79.8%) | [74.5; 84.3] | 1.07 [0.75; 1.53] | 0.702 | 0.97 [0.67; 1.39] | 0.854 |
| | Seasonal | 415 (83.4%) | [79.5; 86.6] | 1.36 [0.97; 1.90] | 0.073 | 1.09 [0.77; 1.55] | 0.604 |
| | All year | 872 (82.8%) | [79.8; 85.4] | 1.3 [0.99; 1.71] | 0.056 | 1.07 [0.80; 1.43] | 0.651 |
| | **Household wealth index** | | | | | | |
| | Poorest | 366 (82.8%) | [78.9; 86.0] | ref | | | |
| | Poorer | 421 (81.6%) | [77.9; 84.8] | 0.92 [0.66; 1.30] | 0.653 | | |
| | Middle | 417 (82.2%) | [78.2; 85.6] | 0.96 [0.67; 1.37] | 0.825 | | |
| | Richer | 422 (81.4%) | [76.7; 85.4] | 0.92 [0.62; 1.34] | 0.637 | | |
| | Richest | 446 (79.9%) | [75.6; 83.6] | 0.83 [0.58; 1.18] | 0.292 | | |
| | **Owns health insurance** | | | | | | |
| | No | 2040 (81.6%) | [79.8; 83.3] | ref | | | |
| | Yes | 32 (74.6%) | [49.3; 89.9] | 0.66 [0.22; 2.00] | 0.461 | | |
| | **Owns mobile phone** | | | | | | |
| | No | 447 (81.7%) | [77.9; 84.9] | ref | | | |
| | Yes | 1625 (81.5%) | [79.4; 83.4] | 0.99 [0.76; 1.28] | 0.917 | | |
| | **Issue perceived as a big problem to seek care for self** | | | | | | |
| | Distance to health facility | | | | | | |
| | Not a problem | 1340 (82.1%) | [79.8; 84.2] | ref | | | |
| | Big problem | 732 (80.3%) | [77.3; 83.0] | 0.89 [0.71; 1.12] | 0.307 | | |
| | Getting permission to go | | | | | | |
| | Not a problem | 1612 (82.3%) | [80.3; 84.1] | ref | | ref | |
| | Big problem | 460 (78.7%) | [74.2; 82.6] | 0.79 [0.61; 1.04] | 0.090 | 0.89 [0.66; 1.21] | 0.473 |
| | Getting money needed for treatment | | | | | | |
| | Not a problem | 917 (82.1%) | [79.6; 84.4] | ref | | | |
| | Big problem | 1155 (81.0%) | [78.6; 83.2] | 0.93 [0.76; 1.14] | 0.494 | | |
| | Not wanting to go alone | | | | | | |
| | Not a problem | 1605 (82.4%) | [80.3; 84.3] | ref | | ref | |
| | Big problem | 467 (78.4%) | [74.4; 82.0] | 0.78 [0.60; 1.00] | 0.052 | 0.95 [0.71; 1.28] | 0.737 |

(*Continued*)

**Table 3.** (Continued)

| | Characteristics | | Early discharge <6 hours | | | | |
| --- | --- | --- | --- | --- | --- | --- | --- |
| | | n (%) | [95%CI] | cOR [95%CI] | p-value | aOR [95%CI] | p-value |
| Women's needs and obstetrics history | **Parity at index birth** | | | | | | |
| | Primiparous | 470 (78.8%) | [75.3; 82.5] | ref | | ref | |
| | Multiparous 2–3 | 789 (82.8%) | [79.9;85.4] | 1.29 [0.99; 1.69] | 0.06 | 1.18 [0.87; 1.61] | 0.269 |
| | Multiparous 4 or more | 813 (81.9%) | [79.2; 84.0] | 1.21 [0.94; 1.57] | 0.145 | 1.04 [0.77; 1.4] | 0.782 |
| | **ANC visits during pregnancy** | | | | | | |
| | None | 68 (83.5%) | [73.0; 90.5] | ref | | | |
| | 1–3 visits | 1060 (82.4%) | [79.8; 84.8] | 0.93 [0.48; 1.77] | 0.818 | | |
| | 4 or more visits | 944 (80.4%) | [77.8; 82.7] | 0.81 [0.42; 1.54] | 0.516 | | |
| | **Timing of first ANC visit** | | | | | | |
| | None | 68 (83.5%) | [73.0; 90.5] | ref | | ref | |
| | During $1^{st}$ trimester | 686 (77.9%) | [74.7; 80.8] | 0.70 [0.37; 1.32] | 0.266 | 0.65 [0.35; 1.19] | 0.163 |
| | Beyond $1^{st}$ trimester | 1318 (83.4%) | [81.2; 85.4] | 0.99 [0.52; 1.89] | 0.975 | 0.87 [0.47; 1.62] | 0.669 |
| | **Birth multiplicity** | | | | | | |
| | Singleton birth | 2021 (81.8%) | | ref | | ref | |
| | Multiple birth | 51 (72.5%) | [60.9; 81.7] | 0.59 [0.34; 1.01] | 0.055 | 0.54 [0.31; 0.94] | 0.030* |
| | **Pregnancy wanted at the time** | | | | | | |
| | Not wanted or wanted later | 304 (76.6%) | | ref | | ref | |
| | Wanted at the time of pregnancy | 1768 (82.4%) | [80.5; 84.2] | 1.43 [1.10; 1.87] | 0.008 | 1.25 [0.95; 1.65] | 0.117 |
| | **Ever had a terminated pregnancy** | | | | | | |
| | No | 1869 (82.9%) | [80.1; 83.7] | ref | | ref | |
| | Yes | 203 (77.6%) | [71.8; 82.4] | 0.76 [0.56; 1.04] | 0.090 | 0.79 [0.57; 1.09] | 0.150 |
| Newborn characteristics | **Newborn sex** | | | | | | |
| | Girl | 1020 (83.0%) | [80.5; 85.2] | 1.21 [0.98; 1.48] | 0.075 | 1.2 [0.57; 1.09] | 0.150 |
| | Boy | 1052 (80.1%) | [77.7; 82.4] | ref | | ref | |
| | **Perceived size at birth** | | | | | | |
| | Smaller than average | 159 (78.1%) | [71.4; 83.6] | ref | | ref | |
| | Average or larger | 1903 (82.0%) | [80.1; 83.7] | 1.28 [0.88; 1.85] | 0.195 | 1.26 [0.89; 1.8] | 0.195 |
| | **Newborn survival** | | | | | | |
| | Survived until survey | 1978 (81.6%) | [79.7; 83.3] | ref | | | |
| | Died on/before discharge | 13 (70.4%) | [48.9; 85.5] | 0.54 [0.21; 1.34] | 0.182 | | |
| | Died after discharge | 81 (82.3%) | [73.5; 88.6] | 1.05 [0.62; 1.77] | 0.849 | | |

*Indicates statistically significant association with the outcome early discharge (p-value<0.05). †Model specification linktest p-value = 0.374; Variance inflation factor = 3.41 (i.e. assumptions met)

Note: ORs <1 refer to lower odds of early discharge compared to the reference category.

vaginally in a health facility. Factors significantly associated (p-value<0.05) with early discharge in binary analysis included region, ethnicity, living situation at the time of the survey, type of facility where index birth occurred, maternal education, timing of first ANC visit, and whether the pregnancy was wanted. Upon adjusting for all factors with p-value<0.2 in the binary logistic regression results, three factors (region, facility type, multiple birth) were significantly associated with discharge <6hours.

Keeping other factors constant, women who gave birth in Conakry (aOR = 3.6 [95% CI = 2.16; 6.01], Faranah (aOR = 1.98 [95%CI = 1.07; 3.68]), Kankan (aOR = 4.04 [95% CI = 2.1; 7.78], Kindia (aOR = 2.83 [1.72; 4.65]) and Nzérékoré (aOR = 4.72 [2.39; 9.34]) had significantly higher odds of discharge <6hours compared to women who gave birth in Boké.

Women who gave birth in a non-government hospital had lower odds of discharge <6hrs (aOR = 0.55 [95%CI = 0.35; 0.85]) compared to those who gave birth in a government lower level facility. Women who had multiple birth had significantly lower odds of discharge <6hrs (aOR = 0.54 [95%CI = 0.31;0.94]) compared to women who had singletons.

## 2. Women who gave birth by caesarean section

Table 4 shows the results of the binary and multivariable logistic regression assessing factors associated with discharge <72hours among women who gave their most recent livebirth via caesarean section. Type of facility where index birth occurred was independently significantly associated (p-value<0.05) with early discharge.

Upon adjusting for factors with p-value<0.2 in the binary logistic regression results, the final model included five factors: region, facility type, household wealth index, parity, and newborn sex. Women who gave birth in a government hospital had significantly lower adjusted odds of discharge <72hours (aOR = 0.09 [95%CI = 0.03; 0.3]) compared to those who gave birth in a government lower-level facility. Women in the richer and richest wealth quintiles had lower odds of early discharge compared to women in the poorest quintile, however not significantly. Multiparous women had higher odds of discharge <72hours compared to primiparous women, however not significantly. Women whose newborn was a girl had lower odds of early discharge (aOR = 0.15 [95%CI = 0.05; 0.48]) compared to those whose newborn was a boy.

## Discussion

This study examined the length-of-stay of women in health facilities postpartum in Guinea. It shows the extremely short durations of stay in health facilities after birth among women who gave birth vaginally, with a median of 3 hours. On the other hand, women who gave birth by caesarean section spent on average more than 5 days in facilities after birth. The analysis highlighted differences in the percentage of women with early discharge depending on international vs. practice-informed cut-offs. The potentially detrimental effects of early discharge on the health and well-being of women could be exacerbated by the absence of a health check from a skilled provider during the short time spent in health facilities. This was particularly noted among women who give vaginal birth, where 30% of those who reported leaving within 2 hours of birth reported not receiving a check. The odds of discharge <6 hours among women who give vaginal birth were higher for women living in certain regions (e.g. Conakry, Kindia, Kankan and Nzérékoré), for those who give birth in non-government lower-level facilities and for those who have a singleton birth. Among women who gave birth by caesarean section, the odds of discharge <72 hours were higher for women who gave birth in government lower-level facilities, and whose newborn was a boy.

Compared to estimates in other countries, our analysis showed that postpartum length-of-stay among women in Guinea are among the shortest globally. Median length-of-stay values of 36 and 252 hours following vaginal and caesarean birth were reported in Cameroon [17]. Kumar and Dhillon [18] reported that in India women spent on average 2.1 an 8.6 days in facilities following vaginal and caesarean births, respectively; and in a rural hospital in Eritrea, Ghebremeskel et al [19] showed a median length-of-stay of 1 day following vaginal birth, and of 6 days following caesarean birth. In Malawi the median length-of-stay was 36 hours after vaginal birth and 180 hours after caesarean section, and in Eswatini, median length-of-stay was 36 hours after vaginal birth and 84 hours after caesarean section [20]. Campbell et al's analysis of 92 countries found that among women with vaginal births, the shortest *mean* length-of-stay was 1.9 days (Uganda 2011) and the lowest *median* was 0.4 days (i.e. ~9.6 hours)

**Table 4. Binary and multivariable† logistic regression assessing factors associated with early discharge (<72 hours, n = 41) among women who gave their most recent livebirth by caesarean section in a health facility in the five years preceding the Guinea DHS2018 (total n = 160).**

| | Characteristics | Early discharge <72 hours | | | | | |
|---|---|---|---|---|---|---|---|
| | | n (%) | [95%CI] | cOR [95%CI] | p-value | aOR [95%CI] | p-value |
| **Community and family factors** | **Region** | | | | | | |
| | Boké | 4 (19.1%) | [6.1; 46.1] | ref | | ref | |
| | Conakry | 9 (26.7%) | [13.4; 46.1] | 1.5 [0.34; 7.2] | 0.583 | 0.99 [0.12; 8.41] | 0.995 |
| | Faranah | 0 (0%) | | 1 | | 1 | |
| | Kankan | 3 (20.1%) | [6.0; 49.9] | 1.1 [0.16; 7.03] | 0.947 | 0.47 [0.04; 5.29] | 0.538 |
| | Kindia | 4 (24.1%) | [8.3; 52.4] | 1.3 [0.22; 8.1] | 0.744 | 1.34 [0.12; 14.8] | 0.807 |
| | Labé | 0 (0%) | | 1 | | 1 | |
| | Mamou | 4 (15.4%) | [4.9; 39.2] | 0.8 [0.12; 4.7] | 0.774 | 0.60 [0.06; 5.79] | 0.659 |
| | Nzérékoré | 17 (67.2%) | [40.9; 85.9] | 8.7 [1.6; 46.9] | 0.013 | 2.88 [0.36; 23.2] | 0.317 |
| | **Residence** | | | | | | |
| | Rural | 10 (25.2%) | [10.8; 48.4] | ref | | | |
| | Urban | 31 (29.5%) | [19.8; 41.5] | 1.2 [0.37; 4.1] | 0.726 | | |
| | **Ethnicity** | | | | | | |
| | Soussou | 7 (21.0%) | [9.6; 39.9] | ref | | | |
| | Peuls | 11 (21.3%) | [11.0; 37.15] | 1.0 [0.3; 3.4] | 0.980 | | |
| | Malinké | 13 (25.3%) | [13.4; 42.6] | 1.3 [0.41; 3.97] | 0.675 | | |
| | Other (Guerzé) | 10 (52.1%) | [26.0; 77.0] | 4.1 [0.91; 18.3] | 0.066 | | |
| | **Living arrangement at time of survey** | | | | | | |
| | Not in union/not living with a partner | 11 (30.1%) | [15.5; 50.3] | ref | | | |
| | Living with a partner | 30 (27.2%) | [18.9; 37.5] | 0.9 [0.36; 2.1] | 0.754 | | |
| | **Involvement in decision making regarding own healthcare** | | | | | | |
| | Not involved | 17 (22.3%) | [13.5; 34.5] | ref | | | |
| | Involved partly or completely | 21 (37.6%) | [23.9; 53.6] | 2.1 [0.87; 5.1] | 0.099 | | |
| | **Number of household members** | | | | | | |
| | 2–3 members | 2 (19.7%) | [3.7; 61.0] | ref | | | |
| | 4–9 members | 24 (26.5%) | [16.3; 39.9] | 1.5 [0.21; 10.1] | 0.695 | | |
| | 10 or more members | 15 (33.7%) | [19.9; 52.1] | 2.1 [0.28; 15.5] | 0.473 | | |
| | **Relation to head of the household** | | | | | | |
| | Self | 3 (30.5%) | [8.9; 66.1] | ref | | | |
| | Partner | 27 (27.8%) | [19.0; 38.7] | 0.9 [0.19; 4.03] | 0.868 | | |
| | Child/child in law | 9 (38.9%) | [20.3; 61.5] | 1.5 [0.35; 6.1] | 0.605 | | |
| | Other | 2 (11.3%) | [2.6; 38.2] | 0.3 [0.03; 2.9] | 0.293 | | |
| **Facility characteristics and norms** | **Type of facility** | | | | | | |
| | Government lower-level facility | 29 (60.6%) | [43.9; 75.1] | | | ref | |
| | Government hospital | 10 (11.9%) | [5.9; 22.3] | 0.09 [0.03; 0.2] | <0.001 | 0.09 [0.03; 0.3] | <0.001* |
| | Non-government lower level facility | 0 (0%) | | 1 | | 1 | |
| | Non-government hospital | 2 (20.3%) | [4.8; 56.5] | 0.17 [0.03; 0.97] | 0.046 | 0.38 [0.03; 5.12] | 0.459 |
| | **Day of birth** | | | | | | |
| | Weekday | 28 (25.3%) | [16.8; 36.2] | 0.6 [0.3; 1.5] | 0.277 | | |
| | Weekend | 13 (35%) | [20.1; 53.5] | ref | | | |

(*Continued*)

**Table 4.** (Continued)

| | Characteristics | Early discharge <72 hours | | | | | |
|---|---|---|---|---|---|---|---|
| | | n (%) | [95%CI] | cOR [95%CI] | p-value | aOR [95%CI] | p-value |
| **Women's socio-economic characteristics** | **Maternal age at birth (in years)** | | | | | | |
| | 13–19 years | 6 (23.5%) | [9.9; 46.3] | ref | | | |
| | 20–24 years | 5 (28.5%) | [12.8; 51.9] | 1.3 [0.3; 5.2] | 0.714 | | |
| | 25–29 years | 10 (25.8%) | [14.1; 42.3] | 1.1 [0.3; 3.6] | 0.839 | | |
| | 30–34 years | 10 (27.6%) | [13.6; 47.9] | 1.2 [0.3; 4.7] | 0.751 | | |
| | 35–49 years | 10 (33.5%) | [18.5; 52.9] | 1.6 [0.4; 5.9] | 0.447 | | |
| | **Highest education level reached** | | | | | | |
| | No education | 26 (32.9%) | [21.5; 46.7] | ref | | | |
| | Primary education | 4 (16.7%) | [5.5; 41.6] | 0.4 [0.1; 1.6] | 0.203 | | |
| | Secondary or higher | 11 (25%) | [13.2; 42.2] | 0.7 [0.3; 1.8] | 0.431 | | |
| | **Occupation frequency** | | | | | | |
| | Not worked in the past 12 months | 16 (34.9%) | [21.0; 51.8] | ref | | | |
| | Occasional | 2 (23.2%) | [4.9; 63.7] | 0.6 [0.08; 3.7] | 0.550 | | |
| | Seasonal | 1 (7.4%) | [1.0; 39.0] | 0.1 [0.01; 1.3] | 0.090 | | |
| | All year | 22 (29.7%) | [18.1; 44.7] | 0.8 [0.31; 1.9] | 0.616 | | |
| | **Household wealth index** | | | | | | |
| | Poorest | 10 (38.9%) | [19.3; 62.9] | ref | | ref | |
| | Poorer | 5 (20.8%) | [7.7; 45.2] | 0.4 [0.09; 1.8] | 0.247 | 0.51 [0.05; 5.1] | 0.561 |
| | Middle | 12 (44.4%) | [24.2; 66.6] | 1.2 [0.32; 4.9] | 0.746 | 1.15 [0.18; 7.2] | 0.877 |
| | Richer | 7 (24.5%) | [11.2; 45.6] | 0.5 [0.13; 2.0] | 0.333 | 0.33 [0.05; 2.4] | 0.272 |
| | Richest | 7 (16.7%) | [7.1; 34.5] | 0.3 [0.07; 1.3] | 0.109 | 0.32 [0.04; 2.6] | 0.282 |
| | **Owns health insurance** | | | | | | |
| | No | 39 (27.5%) | [19.1; 37.8] | ref | | | |
| | Yes | 2 (39.0%) | [8.6; 81.4] | 1.7 [0.24; 12.1] | 0.598 | | |
| | **Owns mobile phone** | | | | | | |
| | No | 2 (22.7%) | [5.82; 58.2] | ref | | | |
| | Yes | 39 (28.5%) | [19.8; 39.1] | 1.3 [0.27; 6.90] | 0.712 | | |
| | **Issue perceived as a big problem to seek care for self** | | | | | | |
| | Distance to health facility | | | | | | |
| | Not a problem | 34 (31.6%) | [21.9; 43.3] | ref | | | |
| | Big problem | 7 (16.2%) | [7.2; 32.4] | 0.4 [0.15; 1.13] | 0.085 | | |
| | Getting permission to go | | | | | | |
| | Not a problem | 34 (29.8%) | [20.5; 41.1] | ref | | | |
| | Big problem | 7 (21.3%) | [9.7; 40.4] | 0.6 [0.23; 1.70] | 0.378 | | |
| | Getting money needed for treatment | | | | | | |
| | Not a problem | 28 (33.7%) | [22.4; 47.1] | ref | | | |
| | Big problem | 13 (18.6%) | [10.6; 30.5] | 0.4 [0.19; 1.00] | 0.058 | | |
| | Not wanting to go alone | | | | | | |
| | Not a problem | 34 (30.5%) | [21.1; 41.8] | ref | | | |
| | Big problem | 7 (17.8%) | [7.7; 36.0] | 0.5 [0.17; 1.40] | 0.183 | | |

(*Continued*)

**Table 4.** (Continued)

| Women's needs and obstetrics history | Characteristics | Early discharge <72 hours | | | | | |
| | | n (%) | [95%CI] | cOR [95%CI] | p-value | aOR [95%CI] | p-value |
|---|---|---|---|---|---|---|---|
| Women's needs and obstetrics history | **Parity at index birth** | | | | | | |
| | Primiparous | 8 (17.0%) | [8.1; 32.4] | ref | | ref | |
| | Multiparous 2–3 | 16 (30.2%) | [18.8; 44.7] | 2.11 [0.78; 5.75] | 0.145 | 2.31 [0.64; 8.4] | 0.200 |
| | Multiparous 4 or more | 17 (34.2%) | [20.5; 51.2] | 2.53 [0.87; 7.37] | 0.087 | 2.82 [0.75; 10.5] | 0.122 |
| | **ANC visits during pregnancy** | | | | | | |
| | None | 1 (17.5%) | [1.9; 69.8] | ref | | | |
| | 1–3 visits | 15 (26.2%) | [15.8; 40.2] | 1.7 [0.14; 19.7] | 0.681 | | |
| | 4 or more visits | 25 (29.8%) | [19.5; 42.7] | 2.0 [0.17; 23.3] | 0.577 | | |
| | **Timing of first ANC visit** | | | | | | |
| | None | 1 (17.5%) | [1.9; 69.8] | ref | | | |
| | During 1st trimester | 10 (19.1%) | [9.5; 34.5] | 1.1 [0.09; 13.8] | 0.080 | | |
| | Beyond 1st trimester | 30 (32.3%) | [21.5; 45.2] | 2.2 [0.19; 25.9] | 0.650 | | |
| | **Birth multiplicity** | | | | | | |
| | Singleton birth | 38 (27.4%) | [18.9; 37.7] | ref | | | |
| | Multiple birth | 3 (46.2%) | [13.7; 82.4] | 2.3 [0.40; 13.2] | 0.354 | | |
| | **Pregnancy wanted at the time** | | | | | | |
| | Not wanted or wanted later | 6 (23.9%) | [9.7; 47.6] | ref | | | |
| | Wanted at the time of pregnancy | 35 (28.7%) | [19.6; 39.9] | 1.3 [0.39; 4.10] | 0.676 | | |
| | **Ever had a terminated pregnancy** | | | | | | |
| | No | 34 (29.6%) | [20.3; 41.0] | ref | | | |
| | Yes | 7 (21.9%) | [9.6; 42.7] | 0.7 [0.23; 1.96] | 0.461 | | |
| Newborn characteristics | **Newborn sex** | | | | | | |
| | Girl | 16 (34.3%) | [12.9; 35.6] | 0.5 [0.25; 1.20] | 0.124 | 0.15 [0.05; 0.48] | 0.001* |
| | Boy | 25 (22.3%) | [23.1; 47.6] | ref | | ref | |
| | **Perceived size at birth** | | | | | | |
| | Smaller than average | 4 (27.1%) | [9.5; 56.9] | ref | | | |
| | Average or larger | 37 (28.2%) | [19.4; 39.2] | 1.1 [0.27; 4.15] | 0.936 | | |
| | **Newborn survival** | | | | 0.952 | | |
| | Survived until survey | 38 (27.9%) | [19.9; 37.6] | ref | | | |
| | Died on /before discharge | 1 (32.1%) | [4.4; 82.9] | 1.2 [0.13; 11.6] | 0.862 | | |
| | Died after discharge | 2 (23.1%) | [4.4; 66.3] | 0.8 [0.12; 5.20] | 0.794 | | |

*Indicates statistically significant association with the outcome early discharge (p-value<0.05). †Model specification linktest p-value = 0.545; Variance inflation factor = 3.42 (i.e. assumptions met)

Note: ORs <1 refer to lower odds of early discharge compared to the reference category.

in Pakistan 2006–7 [12]. Guinea had the highest percentage of early discharge according to the international WHO recommendation, exceeding the 83.2% discharged <24 hours after vaginal birth in Egypt 2008 [12]. Percentage of early discharge among women who had a vaginal birth was documented at 55% in Ghana in 2017 [21], 29.7% in Cameroon [17], 22.3% in India [18], 22% in Eswatini and 10% in Malawi [20]. In India, 12.8% of women who had a caesarean birth were discharged within 72 hours of birth [18], and 15.1% in Cameroon [17]. Discharge <72 hours following caesarean birth was documented among 75.3% of women who gave birth by caesarean section in Egypt 2008 [12], which greatly exceeds that of Guinea.

In comparison to the published literature, evidence shows that factors associated with early discharge are contextual as they vary widely between countries, and our analysis confirms this variability. Considering that this analysis was guided by Campbell et al's conceptual framework [12], factors significantly associated with early discharge in Guinea fell within that framework; however, not all factors in the framework were significant in the Guinean context (e.g. women's age and highest education level, and child survival). Regional variations in length-of-stay was similarly documented in Cameroon [17], however urban vs rural place of residence did not show a significant association in Guinea; this could be possibly explained by a misclassification bias of areas in the DHS survey between urban and rural, a classification that has been described to be too simplistic and fails to document urbanicity on a spectrum [35]. The type of facility was associated with early discharge, in Cameroon same as in Guinea, with government lower-level facilities having higher odds of early discharge [17]. This could be a result of the case-mix in these types of facilities, which usually consists of lesser proportion of women who suffer complications. Another explanation could be the absence or the lack of application of a systematic discharge process in some health centers and health posts, making it easier for women to discharge themselves at the time that they feel is good enough for them to leave.

In several studies, maternal age was associated with early discharge [12, 17, 18, 21], however this was not the case for Guinea showing that women of all ages are being discharged too early, regardless of their varying levels of needs. Adolescent mothers (16% of the sample) might be at higher risks of complications and specific needs for support following childbirth [36, 37]. Similar to Tsiga-Ahmed [20] and Campbell et al. [12], women who had multiple births had lower odds of early discharge in Guinea, while Fouogue [17] and Ghebremeskel [19] found no significant association. Newborn smaller size is a significant determinant of longer length-of-stay [15, 20, 21], however not significantly in our study which could be linked to a reporting bias or lack of accuracy of the used variable. Mothers of newborns who died on or before discharge were less likely to be discharged early from the hospital, which is a factor also observed in Malawi and Eswatini [20], but not significantly in our sample, possibly because of sample size limitations and rare occurrence/reporting of newborn death.

Our findings show that in the period 2013–2018, women who gave birth by caesarean section experienced long stays in health facilities in Guinea, considerably longer than those who gave vaginal birth, and exceeding the global and locally-informed recommendations of length-of-stay. This could be meant to provide support and health monitoring to women during their recovery from the surgery, avoid undetected complications that onset after discharge, provide wound care and remove dressings prior to discharge, thus avoiding the burden of an additional visit/travel to the health facility for dressing removal. This practice takes in consideration the low proportion of women who seek an outpatient postnatal check after discharge [25], and ensures that women with high-risk (post-surgery) receive the care they need without having to make a second trip to the facility considering transport and distance related challenges [28]. Another explanation could be due to maternal or newborn complications that necessitated longer stays, such as post caesarean section surgical site infection requiring longer hospitalisation. Despite a declining trend in post caesarean section infections in Guinea following the establishment of infection prevention measures linked to the Ebola epidemic, their occurrence remains frequent at 5% in 2015 [38]. Detention for inability to cover healthcare fees has been described in other settings as a possible explanation for length-of-stays that are longer than the recommendations [39]. In Guinea childbirth care is nominally free of charge for both vaginal and caesarean births [31], however women might still be asked to make certain informal payments; whether women who fail to make them would be retained is unknown and should be explored further.

Length-of-stay following vaginal birth in Guinea is shorter than the internationally and locally recommended standard, with discharge <24 hours being almost universal in this nationally-representative sample. Even after adapting the cut-offs of early discharge to reflect Guinean resource-informed postpartum care norms, the percentage of early discharge remains above 80%. These short stays render vaginal childbirth comparable to an outpatient service. In some high-income countries, such as Austria and the Netherlands, same day discharge is an option available for women who experience vaginal birth with no complications; yet it is accompanied by regular home visits from a midwife, scheduled follow-ups with a paediatrician, and ensuring easy access to a health facility [40]. Such follow-up model is not officially described, recommended, provided or financed in the Guinean guideline on postpartum care, thus risking that the majority of women leave care environments without sufficient monitoring and support during the critical immediate postpartum period, and without receiving adequate discharge preparedness [5]. This point is further demonstrated by the high percentage of women who report being discharged within 2 hours of birth without receiving a health check (30%) and those who report not receiving an outpatient postpartum care check beyond the first 24 hours after birth (43%) [25].

The results of the regression models reflect that factors associated with early discharge were mostly community and facility related characteristics, which warrants a health systems understanding of the issue. This suggests that the highly common short length-of-stays reflect a variability in facility norms of postpartum care provision and in the application of the national-level guidelines. It is possible that this is in part due to barriers at various levels of the system [10], facility and community norms around immediate postpartum care, and resource availability and training of care providers [15, 22, 36]. These include examples of women leaving the facility early as a result of lack of bed availability, lack of attention from care providers due to high workload, unfavourable environment with limited access to water and sanitation facilities, water, food and social support [22]. In Guinea, essential amenities and comfort features (e.g. electricity and clean water) are lacking in most facilities [41], which also experience a critical shortage of skilled care providers [42]. Both Kindia and Kankan, regions with higher odds of early discharge, are among the regions with low percentages of facilities with availability of trained providers and of guidelines for essential childbirth care [41], and where almost half of births are not attended by a skilled provider [25]. Higher odds of early discharge in Nzérékoré and Conakry, where skilled attendance at birth is 96% and 64% (the top two regions in the country) [25] could be explained by the possible overcrowding of facilities, potentially leading to shorter postpartum length-of-stay in order to reduce bed occupancy [22].

Resource (un)availability does not entirely explain the higher odds of early discharge in Conakry and Nzérékoré. Infectious disease outbreaks are known to have indirect effects on the quality of care and to reduce population trust levels in the health system [43]. Globally, research suggests that women were being discharged earlier from facilities after birth during the COVID-19 pandemic to reduce risk of exposure to the virus [44], and these practices are maintained in some facilities/countries post COVID-19. In Guinea, significant reductions in facility-based births and antenatal care visits were documented during the Ebola Virus disease outbreak in 2014–2016 [32]. This outbreak period overlaps with the DHS' recall period which could explain the overall short length-of-stay nationally, with women preferring to leave the health facilities as early as possible in fear of virus transmission. Additionally, the concentration of the disease spread in the Forest region, which includes Nzérékoré [32], could be a potential explanation for the higher odds of early discharge in this particular region in our data.

Individual-level factors associated with early discharge included multiplicity of births for women who had a vaginal birth, and newborn sex for women who had a caesarean section. Women who had multiple births had lower odds of early discharge, which could reflect their

or their babies' needs for greater care or longer time needed for monitoring and recovery [12]. Postpartum women whose newborns were girls had higher odds of staying in the facility the recommended LoS compared to boys, in Guinea similarly to the results of the multi-country analysis [12]. Reasons might vary by context and gender-related social and cultural norms and values given to babies based on sex, which could be materialised in the length of the period of facility stay after birth. It could also be a reflection social norms of son preferences [45] which could manifest differently between settings (e.g. by quickly securing the needed funds to ensure a faster transport back home for boys). Interestingly, the direction of this association was reversed among women who had a vaginal birth and not significant either, which puts our hypothesis on gendernorms into question and warrants further anthropological exploration of the issue.

## Strengths and limitations

This study provided nationally-representative estimates of length-of-stay and early discharge among women who gave birth in health facilities in Guinea between 2013 and 2018. The study's presentation of early discharge with two cut-offs is a strength; the WHO cut-off allows for consistency in reporting and comparability of findings with different settings, and the locally-informed cut-off represents usual norms of practice in local healthcare facilities across Guinea, making the results context-relevant, and showing that global reporting indicators do not always reflect local realities.

Nonetheless this paper has some limitations, several of which are linked to the unavailability of information in DHS on relevant topics, including: maternal health outcomes which are important determinants of length-of-stay (e.g. postpartum complications, premature birth) [19, 21]; referral status and how women report length-of-stay when referred; time of birth (day or night) which could play a role in length-of-stay (e.g. longer length-of-stay to avoid discharge during the late hours of night) [19], women's preferences and experience of care, and distance from health facility. Another limitation is that the DHS did not collect data about women whose birth outcome was a stillbirth, which limits the exploration of how immediate postpartum care is provided for this specific sub-group who may have higher levels of needs. It should be noted that the outcome variable, as well as some other self-reported variables (e.g. check before discharge, birth attendant, check performer, and others) could be subject to recall bias given the 5-year recall period of DHS data. There is also a potential missing data bias, as missingness in the outcome variable was not independent of demographic characteristics including region, urban residence, ethnicity, type of facility of birth, education level, occupation, mobile phone ownership, and parity. Information bias is also possible with regards to the outcome, considering some inconsistencies noted in the data, such as women who reported having a caesarean birth with extremely low length-of-stay i.e. discharge within 6 hours of birth (29 women, 17 of them were from Nzérékoré (58%), majority of them had no formal education (75%) and 22/29 gave birth in government health centers. The reported lack of skilled attendance at birth by 6% of women in our sample could be a result of reporting bias or a reflection of human-resource challenges. Excluding women who stayed longer than 3 weeks as outliers could exclude some women who had severely premature newborns or other complications, however we used this cut-off consistently with previous research on the topic [12]. The smaller sample size of women who had a caesarean section could have implications on results of the stratified regression analysis.

## Implications for research and practice

This study has implications for future research, including a need for in-depth understanding of factors that lead to short postpartum length-of-stay through qualitative research, to unpack

woman-informed, health worker-informed, and resource-informed reasons of short facility stays. This also warrants qualitatively exploring drivers of postnatal care guideline implementation in Guinea, especially at lower-level (primary) healthcare facilities and in the private sector. Of interest is also exploring the content, frequency and quality of immediate postnatal care and discharge preparedness protocols and practices. Future research should also explore effects of length-of-stay on maternal and newborn outcomes in low-resource settings. There is a need to strengthen the provision of immediate postnatal care in Guinea, including discharge preparedness, by adopting a health-systems lens and a participatory approach that takes into consideration available resources, care providers' needs, and respects women's and families' choices and preferences in the immediate postpartum period.

## Conclusion

The findings highlight length-of-stays that are too short for women who have a vaginal birth in healthcare facilities in Guinea, including notable regional disparities. Factors determining early discharge were mostly documented at the health system or structural level, suggesting that individual needs play a minimal role in defining length-of-stay. Early discharge is exacerbated by the possibility of receiving sub-optimal content, frequency and quality of immediate postpartum care. This study uncovered important gaps in the provision of immediate postnatal care which should be explored in-depth and prioritised on the decision-making agenda to promote the health and wellbeing of women along the continuum of maternity care in Guinea.

## Supporting information

**S1 Checklist. STROBE_checklist.**
(DOCX)

**S1 Table. Variable definitions.**
(DOCX)

**S2 Table. Outliers.**
(DOCX)

**S3 Table. Missing data.**
(DOCX)

## Acknowledgments

We would like to acknowledge the women who took the time to respond to the DHS.

## Author Contributions

**Conceptualization:** Thérèse Delvaux.

**Data curation:** Aline Semaan.

**Formal analysis:** Aline Semaan.

**Methodology:** Aline Semaan.

**Supervision:** Natasha Housseine, Thomas van den Akker, Alexandre Delamou, Lenka Beňová.

**Visualization:** Aline Semaan.

**Writing – original draft:** Aline Semaan.

**Writing – review & editing:** Fassou Mathias Grovogui, Thérèse Delvaux, Natasha Housseine, Thomas van den Akker, Alexandre Delamou, Lenka Beňová.

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
