## [Decision Letter · Decision Letter 0]

14 Jun 2024

PGPH-D-24-00984

Prevalence of and factors associated with early discharge after birth in health facilities in Guinea by mode of birth: Secondary analysis of Demographic and Health Survey 2018

Dear Dr. Semaan,

Thank you for submitting your manuscript to PLOS Global Public Health. After careful consideration, we feel that it has merit but does not fully meet PLOS Global Public Health’s publication criteria as it currently stands. Therefore, we invite you to submit a revised version of the manuscript that addresses the points raised during the review process.

Please note that we have only been able to secure a single reviewer to assess your manuscript. We are issuing a decision on your manuscript at this point to prevent further delays in the evaluation of your manuscript. Please be aware that the editor who handles your revised manuscript might find it necessary to invite additional reviewers to assess this work once the revised manuscript is submitted. However, we will aim to proceed on the basis of this single review if possible.

The reviewer has requested clarity regarding the overall rationale for this study, and has requested additional details regarding the methodology. Please ensure you address each of the reviewer's comments when revising your manuscript.

We look forward to receiving your revised manuscript.

Kind regards,

Hugh Cowley

Staff Editor

Journal Requirements:

Additional Editor Comments (if provided):

Reviewers' comments:

Reviewer's Responses to Questions

**Comments to the Author**

1. Does this manuscript meet PLOS Global Public Health’s publication criteria? Is the manuscript technically sound, and do the data support the conclusions? The manuscript must describe methodologically and ethically rigorous research with conclusions that are appropriately drawn based on the data presented.

Reviewer #1: Yes

2. Has the statistical analysis been performed appropriately and rigorously?

Reviewer #1: Yes

3. Have the authors made all data underlying the findings in their manuscript fully available (please refer to the Data Availability Statement at the start of the manuscript PDF file)?

Reviewer #1: Yes

4. Is the manuscript presented in an intelligible fashion and written in standard English?

Reviewer #1: Yes

5. Review Comments to the Author

Reviewer #1: Introduction

The introduction could benefit from a clearer structure. Consider breaking it into smaller, titled subsections (e.g., "Global Context", "Local Context", "Study Objectives") to improve readability.

While the introduction mentions the lack of sufficient evidence informing postnatal care models, it could more explicitly highlight specific gaps that this study aims to fill. This would strengthen the rationale for the study.

Some paragraphs transition abruptly. For example, the shift from discussing global guidelines to specific barriers in Guinea could be smoother. Adding transitional sentences can help maintain a logical flow.

Please ensure that all claims are backed by appropriate citations. For instance, when discussing barriers to postnatal care in Guinea, ensure each point is referenced.

Some sentences are long and complex. Breaking them into shorter, clearer sentences would enhance readability.

Methods;

While the handling of missing data is explained, it would be beneficial to provide a more detailed justification for excluding certain data and how it might impact the study’s findings. Sensitivity analyses are mentioned but could be elaborated upon.

The explanation of statistical methods (e.g., logistic regression) is clear, but could be enhanced with more details on how potential confounders are controlled for and how model assumptions are checked.

Please provide more information on how participant confidentiality and data protection were ensured, especially given the secondary nature of the data analysis.

Please ensure all claims are well-supported with references

Results

Use conventional methods of reporting results, e.g Mean (SD), IQR, etc

6. PLOS authors have the option to publish the peer review history of their article (what does this mean?). If published, this will include your full peer review and any attached files.

**Do you want your identity to be public for this peer review?** For information about this choice, including consent withdrawal, please see our Privacy Policy.

Reviewer #1: No

---

## [Decision Letter · Decision Letter 1]

5 Aug 2024

PGPH-D-24-00984R1

Prevalence of and factors associated with early discharge after birth in health facilities in Guinea by mode of birth: Secondary analysis of Demographic and Health Survey 2018

 Dear Dr. Semaan

Thank you for submitting your manuscript to PLOS Global Public Health. After careful consideration, we feel that it has merit but does not fully meet PLOS Global Public Health’s publication criteria as it currently stands. Please note that the second reviewer have raised number of pertinent comments. Therefore, we invite you to submit a revised version of the manuscript that addresses the points raised during the review process. 

We look forward to receiving your revised manuscript.

Kind regards,

Ashish KC

Academic Editor

Journal Requirements:

Additional Editor Comments (if provided):

Reviewers' comments:

Reviewer's Responses to Questions

**Comments to the Author**

1. If the authors have adequately addressed your comments raised in a previous round of review and you feel that this manuscript is now acceptable for publication, you may indicate that here to bypass the “Comments to the Author” section, enter your conflict of interest statement in the “Confidential to Editor” section, and submit your "Accept" recommendation.

Reviewer #1: All comments have been addressed

Reviewer #2: All comments have been addressed

2. Does this manuscript meet PLOS Global Public Health’s publication criteria? Is the manuscript technically sound, and do the data support the conclusions? The manuscript must describe methodologically and ethically rigorous research with conclusions that are appropriately drawn based on the data presented.

Reviewer #1: Yes

Reviewer #2: Yes

3. Has the statistical analysis been performed appropriately and rigorously?

Reviewer #1: Yes

Reviewer #2: No

4. Have the authors made all data underlying the findings in their manuscript fully available (please refer to the Data Availability Statement at the start of the manuscript PDF file)?

Reviewer #1: Yes

Reviewer #2: Yes

5. Is the manuscript presented in an intelligible fashion and written in standard English?

Reviewer #1: Yes

Reviewer #2: Yes

6. Review Comments to the Author

Reviewer #1: None

Reviewer #2: Thank you for inviting me to review this interesting article on assessing the factors associated timing of discharge after birth. I have some major conceptual, methodological and presentation concerns of the work.

Title- The term "Prevalence" is used for any disease or event of concern such as mistreatment or poor quality of care. In this aspect, is the timing of birth considered as event of concern? I would suggest removing the term, in terms of the epidemiological aspect (where, when, why, how and who), it is concerned with the timing of discharge (when).

Abstract-

There can be multi-dimensional factors associated with the timing of discharge. Can the authors be specific with socio-demographic, obstetric and health system factor in the objective.

Can you please keep the method section separate and result section.

In the method section, can you provide the study design? In this section, provide the number of women who were eligible and who were enrolled. I am concerned with the two cut-off out vaginal birth, is it by non-assisted and assisted vaginal birth.

In the result section, can you please provide the odds ratio such as aOR, 1.89 95%CI; XX, XX). I guess the result section require more presentation than the conclusion.

The conclusion on Guniea having the shortest length in comparison with other countries, is not presented, please remove this. Your conclusion should always be supported by results. Also, is short stay a marker of adequate care, this is not either presented or rationalize. Also, your remarks on "the health system and structural factors..." need to have evidence. I suggest removing it.

Main text

Introduction

I am surprised to see in the introduction that the authors have three paragraphs on the global context of postpartum care and time of discharge. This might be reflective of the lead authors from Global North. I think this needs to be shortened. Authors need to provide the context of Guniea and this is of concern. The introduction needs to provide in at least to a good description of the social and health system context for pregnancy, intrapartum and postpartum care. What is the national policy for intrapartum and postpartum care? How many women are delivering at health facilities and what has been the trend since MDG period? You also need to provide postpartum set-up in different public health facilities. I think a paragraph on Guniea neither justify the rationale for work nor the collaboration done to generate new evidence on timing of discharge. Some of the contexts you have mentioned in the study setting should be mentioned in the introduction.

I still have concerns with the term "prevalence" in the objective.

Method

What is the study design. A sub-section of study design and the participant selection criteria for the primary study needs to be mentioned.

The variables are well explained and are commendable.

I am concerned with the analysis of lumping all the variables together. Is there an analytical framework (DAG) or are you doing an exploratory analysis using single multi-variable logistic regression? If I understand this, you want the possible association between three domains (socio-demographic, obstetric and health system)-Multi-level analysis. This concerned is seen in table 3, overadjustment of variables, since using a weaker design such as cross-sectional with over-adjustment can not show a true epidemiological association. https://www.sciencedirect.com/science/article/abs/pii/S0895435622001433

Can you please find some primer for socio-demographic, obstetric and health system. I know VIF gives some, but for too many confounders, VIF might not be appropriate.

7. PLOS authors have the option to publish the peer review history of their article (what does this mean?). If published, this will include your full peer review and any attached files.

**Do you want your identity to be public for this peer review?** For information about this choice, including consent withdrawal, please see our Privacy Policy.

Reviewer #1: No

Reviewer #2: No

---

## [Decision Letter · Decision Letter 2]

30 Aug 2024

PGPH-D-24-00984R2

Length-of-stay and factors associated with early discharge after birth in health facilities in Guinea by mode of birth: Secondary analysis of Demographic and Health Survey 2018

Dear Dr. Semaan,

Thank you for submitting your manuscript to PLOS Global Public Health. After careful consideration, we feel that it has merit but does not fully meet PLOS Global Public Health’s publication criteria as it currently stands. Therefore, we invite you to submit a revised version of the manuscript that addresses the points raised during the review process.

We look forward to receiving your revised manuscript.

Kind regards,

Ashish KC

Academic Editor

Journal Requirements:

Additional Editor Comments (if provided):

Dear Dr. Semaan

We have now received the comments from the reviewer. We would request you to make following changes to the submitted version

Abstract- Provide the sub-heading of Background, method, results and conclusion

Please provide the STROBE checklist and ensure reporting of the manuscript as per the STROBE checklist.

Reviewers' comments:

Reviewer's Responses to Questions

**Comments to the Author**

1. If the authors have adequately addressed your comments raised in a previous round of review and you feel that this manuscript is now acceptable for publication, you may indicate that here to bypass the “Comments to the Author” section, enter your conflict of interest statement in the “Confidential to Editor” section, and submit your "Accept" recommendation.

Reviewer #2: All comments have been addressed

2. Does this manuscript meet PLOS Global Public Health’s publication criteria? Is the manuscript technically sound, and do the data support the conclusions? The manuscript must describe methodologically and ethically rigorous research with conclusions that are appropriately drawn based on the data presented.

Reviewer #2: Yes

3. Has the statistical analysis been performed appropriately and rigorously?

Reviewer #2: Yes

4. Have the authors made all data underlying the findings in their manuscript fully available (please refer to the Data Availability Statement at the start of the manuscript PDF file)?

Reviewer #2: Yes

5. Is the manuscript presented in an intelligible fashion and written in standard English?

Reviewer #2: Yes

6. Review Comments to the Author

Reviewer #2: My comments have been well address, and this will be a good literature for global health practitioners on the length of stay.

Please note that length of stay cannot be termed as prevalence, unless it is considered as a problem or disease exposure/outcome. Reference to publication does not correctly state the standard epidemiology text books. Please refer to Woodward M. Epidemiology: Study Design and Data Analysis. Florida: CRC Press, 2014, 3rd Edition.

7. PLOS authors have the option to publish the peer review history of their article (what does this mean?). If published, this will include your full peer review and any attached files.

**Do you want your identity to be public for this peer review?** For information about this choice, including consent withdrawal, please see our Privacy Policy.

Reviewer #2: No

---

## [Editor Report · Decision Letter 3]

11 Sep 2024

Length-of-stay and factors associated with early discharge after birth in health facilities in Guinea by mode of birth: Secondary analysis of Demographic and Health Survey 2018

PGPH-D-24-00984R3

Dear Ms Semaan,

We are pleased to inform you that your manuscript 'Length-of-stay and factors associated with early discharge after birth in health facilities in Guinea by mode of birth: Secondary analysis of Demographic and Health Survey 2018' has been provisionally accepted for publication in PLOS Global Public Health.

Best regards,

Ashish KC

Academic Editor